# Microplastics in the Lung Tissues Associated with Blood Test Index

**DOI:** 10.3390/toxics11090759

**Published:** 2023-09-06

**Authors:** Shuguang Wang, Wenfeng Lu, Qingdong Cao, Changli Tu, Chenghui Zhong, Lan Qiu, Saifeng Li, Han Zhang, Meiqi Lan, Liqiu Qiu, Xiaoliang Li, Yuewei Liu, Yun Zhou, Jing Liu

**Affiliations:** 1State Key Laboratory of Respiratory Disease, The First Affiliated Hospital of Guangzhou Medical University, Guangzhou 510120, China; wangshuguang1996@163.com (S.W.); karsawithl@163.com (W.L.); 2School of Public Health, Guangzhou Medical University, Guangzhou 511436, China; zchenghui04@gmail.com (C.Z.); lane_q@126.com (L.Q.); 18124597753@163.com (S.L.); zhanghan233326@stu.gzhmu.edu.cn (H.Z.); 2022210587@stu.gzhmu.edu.cn (M.L.); qliqiu@stu.gzhmu.edu.cn (L.Q.); 3Department of Thoracic Surgery, The Fifth Affiliated Hospital, Sun Yat-sen Unversity, Zhuhai 519000, China; caoqd@mail.sysu.edu.cn; 4Department of Pulmonary and Critical Care Medicine, The Fifth Affiliated Hospital, Sun Yat-sen University, Guangzhou 519000, China; tuchangli2008@126.com; 5Zhuhai Center for Chronic Disease Control and Prevention, Zhuhai 519060, China; lxlstudent@163.com; 6Department of Epidemiology, School of Public Health, Sun Yat-sen University, Guangzhou 510080, China; liuyuewei@mail.sysu.edu.cn

**Keywords:** MPs, lung tissue, quantitative assessment, internal exposure, health effects

## Abstract

Microplastics (MPs) have received a lot of attention and have been detected in multiple environmental matrices as a new environmental hazard, but studies on human internal exposure to MPs are limited. Here, we collected lung tissue samples from 12 nonsmoking patients to evaluate the characteristics of MPs in human lung tissues using an Agilent 8700 laser infrared imaging spectrometer and scanning electron microscopy. We detected 108 MPs covering 12 types in the lung tissue samples, with a median concentration of 2.19 particles/g. Most of the MPs (88.89%) were sized between 20 to 100 μm. Polypropylene accounts for 34.26% of the MPs in the lung tissues, followed by polyethylene terephthalate (21.30%) and polystyrene (8.33%). Compared with males and those living far from a major road (≥300 m), females and those living near the main road (<300 m) had higher levels of MPs in lung tissues, which positively correlated with platelet (PLT), thrombocytocrit, fibrinogen (FIB), and negatively related with direct bilirubin (DB). These findings help confirm the presence in the respiratory system and suggest the potential sources and health effects of inhaled MPs.

## 1. Introduction

Microplastics (MPs) are plastic particles, debris, fragments, or fibers less than 5 mm in size, which was first proposed by Thompson et al. [1]. It has attracted worldwide attention in recent years due to the uncontrolled release of over 3 million tons of primary MPs into the global environment annually [2]. MPs were detected in the air [3,4,5], food [6,7,8,9,10], condiments [11,12,13], drinking water [14,15,16,17] and personal care products [18,19], suggesting the potential impact on human health. Inhalation is one of the main routes of internal exposure to MPs [20], which might come from MPs-contaminated air (i.e., wearing face masks [21], occupational MP exposures [22] and cooking processes [23], but the evidence on the presence of MPs in the respiratory system is still limited and inconsistent. The characteristics of MPs may vary in different regions of the respiratory system or different biological samples. For example, the size of MPs was reported between 101 to 301 μm in sputum [24], 1.73 ± 0.15 mm in bronchoalveolar lavage fluid (BALF) [25], and 12 to 16.8 μm in lung tissues [26,27]. In comparison to other biological samples, MPs in the lung tissues provide direct evidence for the deposition of MPs in the respiratory system and suggest the health hazards of MPs [28]. However, the two limited studies reported relatively large differences in the size of MPs, which may be attributable to different detection methods. Most previous studies used traditional techniques such as Raman spectroscopy and Fourier-transform infrared (FTIR) spectroscopy, which generally depend on manual selection and measurement and may increase the bias [29,30]. A more objective and quantitative assessment of the internal dose of MPs is necessary for the exploration of the source and human health effects of MPs.

Therefore, we collected the lung tissue samples from patients hospitalized in the Fifth Affiliated Hospital of Sun Yat-sen University and assessed the characteristics of MPs by an accurate and automated Agilent’s novel Laser Direct Infrared (LDIR) imaging system [31,32] combined with a scanning electron microscope (SEM). We then correlated MPs exposure level with the blood test index. We aimed to provide more evidence on deposited MPs in the respiratory system and reveal the potential health effects of inhaled MPs.

## 2. Methods

### 2.1. Recruitment of Patients and Lung Tissue Collection

In this study, we recruited nonsmoking patients who were finally diagnosed with lung cancer (International Classification of Diseases, Tenth Revision (ICD–10): C34) and hospitalized at the Fifth Affiliated Hospital of Sun Yat-sen University. Information on demographics, lifestyles and occupational history was collected by standardized questionnaires, including sex, age, working indoors, wearing of face masks, alcohol consumption, educational level, BMI level, seafood consumption, traffic pollution exposure time, self-cooking and distance between residence and nearest major roads. Non-smokers were defined as those who smoked less than one cigarette per day during the past six months [33]. To ensure biological safety, patients who had tuberculosis (ICD–10: A15–A19), viral hepatitis (ICD–0: B15–B19) and human immunodeficiency virus diseases (ICD–10: B20–B24) were excluded from this study. A total of 12 nonsmoking patients were included in the final analysis. All the participants matched the indications for a video-assisted thoracoscopic surgery (VATS) lobectomy, including the following: (1) a benign condition that calls for anatomical excision; (2) a malignant condition with tumors less than 6 cm; (3) localization in the periphery or greater than 1 cm from a fissure or greater than 3 cm from the lobar carina; (4) TNM stage I or II; (5) N2 lymph nodes without metastatic involvement; and (6) a single extrapulmonary cancer metastasis that cannot be excised with standard wedge resection [34]. Lung tissue samples were collected by VATS and using metal surgical operation instruments in a clean operation room. Lung tissue samples were taken from normal tissues that were more than 5 cm distant from the lesion site. The samples were placed into metal containers and sectioned by a metal bistoury. All the samples were stored in glass bottles at −80 °C until detection. The informed consent of each participant was obtained before the research procedures started. This study was conducted according to the revised Declaration of Helsinki and modified on the basis of the original ethical document (ID: 2018–K51–1). The ethical approval revision document was approved by the Ethics Committee of the Fifth Affiliated Hospital of Sun Yat-sen University and signed in 2020–11–21.

### 2.2. Pretreatment and Detection of MPs in the Lung Tissues

Lung tissue samples were pretreated in the laboratory to digest natural organics. Each lung tissue sample was weighed in a 100 mL beaker (Huaou Industrial Co., Jiangsu, China) and filled with about three times the amount of concentrated nitric acid (68%, GR; Sinopharm Chemical Reagent Co., Shanghai, China). After being placed at room temperature (25 °C) for 2 days, the sample was stirred and heated for 3 hours (95 ℃) to thoroughly digest the protein. Then, the solution was filtered with a steel membrane (Youmi Industrial Co., Chongqing, China) with a mesh size of 1,000 mesh and a pore size of 13 μm using a vacuum pump. Membranes were rinsed several times with ethanol (AR; Shanghai Titan Technology Co., Shanghai, China) after being rinsed with ultra-pure water. Thirdly, we put the membrane into a 20 mL glass bottle and added sufficient ethanol. The substance on the membrane was dispersed into ethanol by ultrasonic (40 KHz, Shenzhen Jiemeng Technology Co., Guangdong, China) treatment for more than 30 min. Lastly, the membrane was rinsed several times with ethanol. Then, the ethanol solution was concentrated to 100 μL in an infrared rapid drying oven (Hangzhou Qiwei Instrument Co., Zhejiang, China) and dripped to the highly reflective slide (Agilent Technologies Co., Santa Clara, CA, USA). Agilent’s novel Laser Direct Infrared imaging system (LDIR, Agilent 8700, Agilent Technologies Co., Santa Clara, CA, USA) was used to detect MPs after the alcohol was evaporated. Particle analysis module, reflection mode and automatic test method were selected to detect MPs (detection limit: 20–500 μm). Specific types of MPs were distinguished by comparing them with the MPs spectrum library (Shanghai Microspectral Testing Technology Group Co., Shanghai, China). The comparison results returned a matching degree ranging from 0 to 1, indicating the level of resemblance between the sample and the standard. A higher matching degree implies greater reliability of the microplastic. We included MPs with a matching degree ≥ 0.80 in this study. We also excluded polyamide from our study because LDIR cannot distinguish polyamide and protein apart due to their similarity in the spectrum. The concentrations of MPs (particles/g) were calculated by dividing the quantity by the weight of the relevant lung tissue or control sample. We defined fiber as a MP with a length-to-diameter ratio ≥ 3, while the rest of the MPs were defined as irregular particles. Certain MPs were then marked on the highly reflective slides and used scanning electron microscopy (SEM, FEI O45, Thermo Scientific, Eugene, OR, USA) to further analysis on the characteristics of the MPs.

### 2.3. Quality Control

We use plastic-free methods throughout the procedure to minimize plastic contamination, including the use of metal surgical instruments during the procedure, cotton white coats and clean nitrile gloves, as well as glassware for tissue sampling and pretreatment. The metal surgical instruments and glass containers were previously washed with ultrapure water and then dried in an oven. All solutions (e.g., nitric acid, alcohol, and ultrapure water) used in sampling and measuring were filtered through a filter membrane (1 μm pore size) prior to use.

In addition, we performed three blank control samples to cover all potential contamination sources, which were obtained by performing the same steps using 5.0 g ethanol instead of the human lung [35]. The approach for the control group was substantially the same as for the lung tissue group, except that we utilized ethanol instead of human lungs. The opening time of glass bottles containing ethanol was the same as that of tissue samples. The operating rooms were previously cleaned, and conditions remained similar in each operating room during all sampling periods.

To minimize contamination from the surrounding air, precautions were taken during the analysis of the samples. For instance, slides containing concentrated ethanol solution were securely stored in closed Petri dishes, only being opened after complete evaporation of the ethanol. The entire procedure was performed within a meticulously cleaned fume cupboard, with the power turned off and the shield down to reduce unfiltered airflow. To prevent the entry of individuals and minimize airborne contamination from the external environment, the laboratory door remained closed throughout the sample processing. Additionally, each tissue sample was handled separately to avoid any potential cross-contamination. We also used three standard MP samples including polystyrene (PS), polypropylene (PP) and polyethylene (PE) to calculate the recovery rate, which has been described in our previous study [36]. All experiments were repeated independently three times. The average recoveries of PS, PP and PE were 92%, 89% and 78%, respectively, indicating the feasibility of LDIR.

### 2.4. Blood Index Tests

Each participant provided a total of 10 mL of fasting blood, which was divided into one 5 mL ethylenediamine tetraacetic acid (EDTA) anticoagulation tube and one 5 mL coagulation tube for the serum to measure hematological and serum biochemical indices, respectively. Determination of peripheral blood cells by XN-20[A1] fully automatic hematology analyzer (XN-20[A1], Sysmex Corp., Hyogo, Japan), while coagulation parameters were measured by blood coagulation analyzer with magnetic beads (STA-COMPACT, Stago, Paris, France). Liver and kidney function tests and serum electrolytes were measured by colorimetric analysis (HITACHI 7180 (ISE), HITACHI High-Tech Co., Tokyo, Japan).

### 2.5. Statistical Analysis

We used the Mann–Whitney U test to examine the difference in MP concentrations between the lung tissue group and the control groups, as well as the differences in MP exposure concentrations by individual characteristics and the differences in blood test index between low and high MP groups which were divided according to the median concentrations of MPs in the lung tissues (2.19 particles/g). Spearman’s correlation coefficient was used to evaluate the correlation between MP concentration in lung tissues and certain blood test indexes. The absolute value of the correlation coefficient r reflects the correlation between the MP concentrations and these clinical examination indexes, which were classified as quite weak (0.1–0.19), weak (0.20–0.39), moderate (0.40–0.59), strong (0.60–0.79) and extremely strong (0.80–1.00) [37,38]. All statistical analyses were performed using R software version 4.1.1 (R core team, Vienna, Austria) and SPSS26.0 (SPSS Inc., Chicago, IL, USA), and *p*-value < 0.05 was regarded as statistically significant for all two-sided statistical tests.

## 3. Results

### 3.1. Characteristics of MPs in the Lung Tissues

The number and proportion of MPs in each sample are shown in Figure 1. MPs were detected in all samples except for lung tissue sample 9 and control sample 1. We detected 108 MPs including 19 fibers (17.59%) and covering 12 types (Figure 1A,B, and Appendix A) in the 11 lung tissue samples, while 7 MPs of 4 types were in the control samples. The concentration of the total MPs in the lung tissue group was significantly higher than that in the control group (mean [SD]: 4.31 [5.11] and 0.47 [0.50] particles/g, respectively, *p* = 0.04, shown in Table 1). The most abundant MP in the lung tissues was polypropylene (PP, 34.26%), followed by polyethylene terephthalate (PET, 21.30%), polystyrene (PS, 8.33%), polyvinylchloride (PVC, 6.48%), polytetrafluoroethylene (PTFE, 6.48%), chlorinated polyethylene (CPE, 5.56%), polyethylene (PE, 4.63%), acrylates (ACR, 4.63%), ethylene vinyl acetate (EVA, 2.78%), butadiene rubber (BR, 2.78%), polyurethane (PU, 1.85%) and silicone (SIL, 0.92%) (Figure 1C). The majority (88.89%) of MPs were between 20 and 100 μm in diameter, with a median value of 41.36 (IQR, interquartile range: 29.03) μm (Figure 1D). We further characterized the abundant MPs which accounted for more than half of the total MPs (i.e., PP and PET) using SEM (Figure 2). SEM and LDIR were used to offer more information on the surface textures. SEM pictures revealed the surface roughness of several MPs, indicating that different shapes and sizes of MPs were discovered in lung tissues and the detected MPs with signs of weathering.

### 3.2. Concentrations of MPs in the Subgroups

The 12 patients were aged from 29 to 69 years and included 7 males and 5 females. All of them did not have a history of lung or upper airway surgeries (Appendix A). We further performed subgroup analysis based on the individual characteristics (Table 2). Females (Median [IQR]: 7.77 [9.90] particles/g) and those living near major roads (5.21 [5.98] particles/g) had higher concentrations of MPs in the lung tissues when compared with males (1.20 [1.80] particles/g) and those living far from a major road (1.20 [1.50] particles/g). No significant differences in the concentrations of MPs were observed between other individual characteristics including age, working indoors, wearing face masks, alcohol consumption, educational level, BMI level, seafood consumption, traffic pollution exposure time or self-cooking.

### 3.3. Relationships between MPs in the Lung Tissues and Blood Test Index

We compared the difference in the levels of blood test index between low and high MPs groups. Compared with the low MPs group (<2.19 particles/g), the high MPs group (≥2.19 particles/g) had significantly higher levels of platelet (PLT), thrombocytocrit, fibrinogen (FIB), Na^+^ and lower direct bilirubin (DB) (Table 3). Spearman correlation test showed very strong positive correlations in MPs concentrations with thrombocytocrit (r = 0.82, *p* < 0.01), and strongly positive correlations with PLT (r = 0.78, *p* < 0.01), as well as FIB (r = 0.63, *p* = 0.03), while strongly negative correlations with DB (r = −0.78, *p* < 0.01), TB (r = −0.66, *p* = 0.02) and hemobilirubin (r = −0.62, *p* = 0.03, shown in Appendix A).

## 4. Discussion

To our knowledge, this is the first study to objectively and quantitively assess the associations between the MPs in the lung tissues and blood test index in humans. The particle size of MPs was primarily in the range of 20–100 μm. Females and those residing near the main road (<300 m) exhibited higher concentrations of MPs in lung tissues, which favorably correlated with PLT, thrombocytocrit and FIB, and were negatively associated with DB.

Limited studies have reported the presence of MPs in the respiratory system. The characteristics of MPs especially the size may vary in different regions of the respiratory system. Huang et al. employed LDIR and FTIR to identify MPs in sputum (upper airways) and discovered most MPs with sizes ranging from 101 to 301 μm [24]. Using Raman spectroscopy, Amato-Lourenço et al. collected the lung tissues from 20 decedents and discovered that the size of MPs in the lung tissues was between 1.60 and 16.80 μm [26]. In our study, we used LDIR combined with SEM to detect the MPs and observed a main size range of 20–100 μm in the lung tissues. These outcomes are consistent with our assumptions about the association between airway features and size-fractioned MPs that the smaller-sized MPs could enter and deposit in the deeper respiratory tract. We also noted that the size of detected MPs in our study was larger than that reported by Amato-Lourenço et al. [26]. One possible reason is that the MP particle size distribution may be different between decedents and living participants. It was between 1.60 and 16.80 μm in decedents while much larger in living persons (12 to 2,475 μm) [27], which was similar to our findings. Another reason may be different detection methods. We did not discover MPs with a particle size smaller than 20 μm owing to the limitation of the detection quadrant by LDIR (20–500 μm), while they discovered smaller MPs by Raman spectroscopy or FTIR spectroscopy. Compared with objective LDIR, Raman spectroscopy and FTIR spectroscopy depend on manually selecting and counting, resulting in an artificial selection bias where certain suspected MPs are easily missed or overlooked. In the future, more precise and automated detection methods are needed to comprehensively study MPs in lung tissue. Taken together, these findings indicate a wide range of MPs in size might enter the lower respiratory tract.

Although there are no epidemiological studies on the associations between exposures to MPs by inhalation and blood test index, a recent study indicated that inhaling PS nanoplastics altered a variety of hematologic and biochemical indicators in rats [39]. The blood test index is a common and important indicator of body health. We found that patients in the high MP group have higher levels of PLT, thrombocytocrit and FIB. Meanwhile, it presented a strong correlation between MPs in the tissues and these indices. PLT, thrombocytocrit and FIB are common indices that participate in coagulation by releasing thromboplastin molecules and promoting the formation of prothrombin [40], which are often used to reflect the risk of thrombus-related diseases in clinic. Our research showed that exposure to MPs may lead to peripheral thrombosis. A previous study also showed that PS can enter systemic circulation through the lung and promote peripheral thrombosis in hamsters [41]. Meanwhile, we discovered a negative correlation between the concentration of MPs and DB, which is an indicator of liver injury. A recent study indicated that PS could damage the liver by inflammation, apoptosis and oxidative stress in liver cells [42], which further suggested the damage of exposure to MPs. These results suggested the potential health effects of the inhaled MPs.

Inhalation is one of the main routes of internal exposure to MPs [20], which might come from MPs-contaminated air released by tire wear, brake pad wear, aged traffic signs [43,44,45,46] and washing clothing [47], which are regarded as primary source of air MPs. Previous sputum research discovered a correlation between the quantities of MP types and smoking and suggested that smokers might be exposed to more MPs [24]. In our study, we excluded smokers and discovered amounts of MPs in lung tissues among never smokers, which indicated that MPs may deep into the human respiratory system through a nonsmoking pathway. We also discovered higher concentrations of MPs in the lung tissues of females and those living near major roads which were similar to the results reported by Baeza-Martinez et al. [25]. Previous research has yielded inconclusive results regarding the potential impact of different sexes on the distribution of MPs in the human airway. Jenner et al. discovered a significant difference in the number of MPs, with males exhibiting higher levels [27], whereas Baeza-Martínez et al. found elevated concentrations among females [25]. Conversely, Huang et al. found no significant differences between different sexes in sputum [24]. Further studies are required to comprehensively investigate the factors influencing MP distribution in the human respiratory system. These findings showed that MPs emitted by traffic and washing processes might be potential sources of lung tissue exposure to MPS in non-smokers, emphasizing that MPs-contaminated air exposures could be preventable, especially the traffic pollution source.

Studies have shown that fibers are the dominant found MPs shape in the atmosphere in both indoor and outdoor environments [48,49,50,51]. Fiber MPs are mainly derived from washing products, textiles and tires in our daily life [25]. Previous studies found that the main shape of MPs was fiber in lung tissue [27] and BALF [25]. However, we found that the fiber MPs were at a much lower frequency in the lung tissues, which was similar to the result of Amato-Lourenço et al. [26]. The lower number of fibers observed in this study could be attributed to the fact that only fibers with a physical diameter of less than 3 µm are able to reach the alveolar region [52]. It may also be related to the use of furniture and clothing. We discovered about 4.31 MP particles per gram of lung tissue, which was greater than the 0.56 MP particles per g reported by Amato-Lourenço et al. [26] and similar to the result obtained by Jenner et al. [27]. Given that the average weight of a set of normal adult lungs is around 840 g [53], there are 3620.4 particles in two lungs. It is reported that an adult male inhaled 62,000 MPs annually [54]. These indicate that a large amount of MPs is retained in the upper airway.

This research has several strengths. First, we employed a more accurate and automated LDIR to detect MPs, which could limit the possible bias induced by manual selection and misidentification of MPs, and the shape of MPs was further characterized by SEM. Second, we directly quantified the deposition of MPs in the lung parenchyma. Our research could better reflect the internal exposure level of MPs in human respiratory systems because our samples were from living people and the different positions of lung tissue. Third, we analyzed the potential sources and health effects of the inhaled MPs on the human body. Last, we carried out quality control by establishing procedural blank controls and high matching degrees to reduce the error of misclassifying non-microplastic particles as MPs, resulting in more precise and objective findings.

There were certain limitations to our research. The sample size in this study was relatively small, which limited the assessment of dose–response relationships between MP exposure and blood test index levels, and could not provide enough statistical power to observe the significant difference of the certain MPs between the lung tissue group and control group. In addition, not all MPs in the lung tissue were identified since LDIR could not discriminate polyamides and natural proteins and could not identify MPs beyond 20 to 500 μm.

## 5. Conclusions

This study quantified MPs in lung tissue and suggested that humans inadvertently inhaled MPs. The particle size of MPs was primarily in the range of 20–100 μm. Females and those residing near the main road (<300 m) had higher levels of MPs in lung tissues, which favorably correlated with PLT, thrombocytocrit and FIB, and were negatively associated with DB. These suggested that MPs emitted by traffic and washing processes might be potential sources of MP exposure in lung tissue and revealed a correlation between MP exposure and certain blood test indices. This study gives fresh information on inhaled MPs’ internal exposure and provides fundamental information on the potential health effects of MPs on humans. Further research is needed to validate our findings and elucidate the underlying mechanisms.

## Figures and Tables

**Figure 1 toxics-11-00759-f001:**
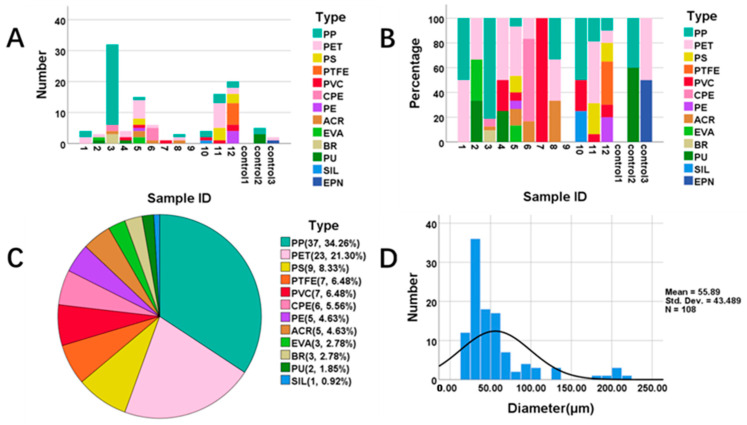
The characteristics of MPs in lung tissues and controls. (**A**,**B**) The number and proportion of MPs in each sample. (**C**) The proportion of different types of MPs in the lung tissue samples. (**D**) The size distributions of MPs in the lung tissue samples. MPs with a matching degree ≥ 0.80 were included. Note: MPs: microplastics, PP: polypropylene, PET: polyethylene terephthalate, PS: polystyrene, PVC: polyvinylchloride, PTFE: polytetrafluoroethylene, CPE: chlorinated polyethylene, PE: polyethylene, ACR: acrylates, EVA: ethylene vinyl acetate, BR: butadiene rubber, PU: polyurethane, SIL: silicone, EPN: phenolic epoxy resin.

**Figure 2 toxics-11-00759-f002:**
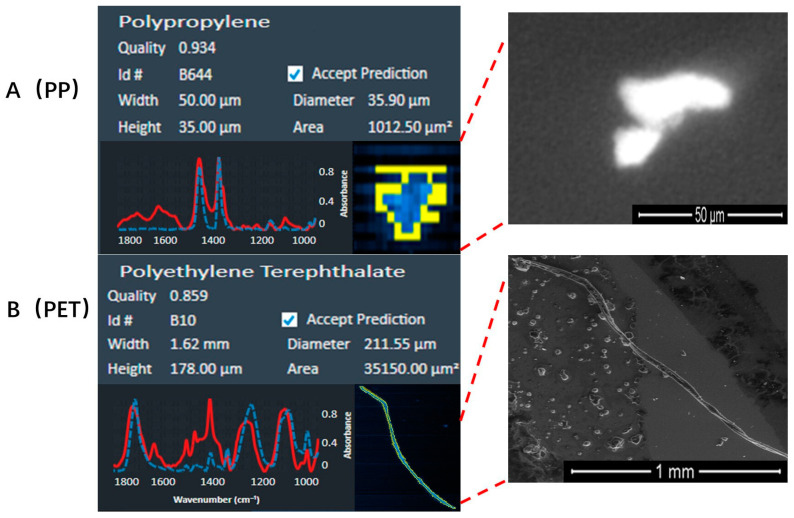
Surface textures of the top two abundant MPs. The types and sizes of MPs identified by LDIR are shown on the left; red solid and blue dotted lines, respectively, indicate the laser infrared spectrum of the sample and standard microplastics. While the electron microscope shows the shape of the MPs on the right. MPs with a matching degree ≥ 0.80 were included. Note: MPs: microplastics, PP: polypropylene, PET: polyethylene terephthalate.

**Table 1 toxics-11-00759-t001:** Comparison of MPs’ concentrations between lung tissue and control group (matching degree ≥ 0.80 ^a^).

MPS	Lung Tissue	Control	*p*-Value ^c^
	Mean (SD), Particles/g	Mean (SD), Particles/g	
Total	4.31 (5.11)	0.47 (0.50)	0.04
PP	1.65 (4.02)	0.13 (0.23)	0.22
PET	0.94 (1.47)	0.07 (0.12)	0.14
PS	0.35 (0.73)	0 ^b^	0.35
PVC	0.25 (0.27)	0 ^b^	0.14
PTFE	0.15 (0.53)	0 ^b^	0.62
CPE	0.26 (0.62)	0 ^b^	0.46
PE	0.13 (0.32)	0 ^b^	0.46
ACR	0.22 (0.35)	0 ^b^	0.27
EVA	0.12 (0.31)	0 ^b^	0.46
BR	0.18 (0.49)	0 ^b^	0.46
PU	0.06 (0.13)	0.20 (0.35)	0.41
SIL	0.05 (0.17)	0 ^b^	0.62
EPN	0 ^b^	0.07 (0.12)	0.05

Note: mean (SD), standard deviation; MPs: microplastics, PP: polypropylene, PET: polyethylene terephthalate, PS: polystyrene, PVC: polyvinylchloride, PTFE: polytetrafluoroethylene, CPE: chlorinated polyethylene, PE: polyethylene, ACR: acrylates, EVA: ethylene vinyl acetate, BR: butadiene rubber, PU: polyurethane, SIL: silicone, EPN: phenolic epoxy resin. ^a^ MPs with a matching degree ≥ 0.80 were included. ^b^ indicates that there were no MPs within the matching degree. ^c^ The *p*-value was estimated using the Mann−Whitney U test.

**Table 2 toxics-11-00759-t002:** Comparisons of the concentrations of MPs stratified by the patients’ characteristics (N = 12, matching degree ≥ 0.80 ^a^).

Characteristics	N (%)	Median (IQR), Particles/g	*p*-Value ^b^
Sex			0.01
Male	7 (58.33)	1.20 (1.80)	
Female	5 (41.67)	7.77 (9.90)	
Age, years			0.87
<55	4 (33.33)	2.49 (6.05)	
≥55	8 (66.67)	1.93 (7.31)	
Working indoors			0.09
No	2 (16.67)	11.40	
Yes	10 (83.33)	1.78 (3.22)	
Wearing face masks, hours per day			0.37
<5.25	7 (58.33)	2.34 (8.41)	
≥5.25	5 (41.67)	1.51 (3.48)	
Alcohol consumption			0.13
Yes	2 (16.67)	0.87	
No	10 (41.67)	2.64 (6.80)	
Educational level			0.09
Middle school or below	4 (33.33)	6.49 (12.69)	
High school or above	8 (66.67)	1.62 (2.12)	
BMI level			0.47
Non-obese, <24 kg/m^2^	5 (41.67)	2.34 (4.43)	
Overweight or obese, ≥24 kg/m^2^	7 (58.33)	1.20 (7.23)	
Seafood consumption, times per week			0.06
≥4	5 (41.67)	1.06 (3.36)	
<4	7 (58.33)	2.94 (7.96)	
Traffic pollution exposure time, minutes per day			0.40
<30	8 (66.67)	3.78 (8.26)	
≥30	4 (33.33)	1.62 (1.62)	
Self-cooking			0.52
No	6 (50.00)	2.00 (7.79)	
Yes	6 (50.00)	2.19 (6.87)	
Distance between residence and nearest major roads, meter			0.04
<300	5 (41.67)	5.21 (5.98)	
≥300	7 (58.33)	1.20 (1.50)	

Note: median (IQR), IQR, interquartile range, MPs: microplastics. ^a^ MPs with a matching degree ≥ 0.80 were included. ^b^ The *p*-value was estimated using the Mann–Whitney U test.

**Table 3 toxics-11-00759-t003:** Comparison of blood test index between low and high MP concentrations (N = 12, matching degree ≥0.80 ^a^).

Blood Test Index	Median (IQR)		*p*-Value ^b^
	Low MP Group (<2.19 Particles/g)	High MP Group (≥2.19 Particles/g)	
WBC (×10^9^/L)	6.46 (0.98)	7.30 (3.68)	0.75
RBC (×10^9^/L)	5.06 (0.85)	4.40(1.42)	0.26
HGB (g/L)	152.00 (23.00)	136.50 (38.75)	0.17
PLT (×10^9^/L)	187.00 (19.75)	244.50 (119.00)	<0.01
Hematocrit (%)	45.30 (6.23)	40.05 (10.38)	0.20
MCV (fL)	89.55 (2.60)	88.90 (5.73)	1.00
Mean red cell hemoglobin content (pg)	30.25 (0.82)	29.05 (2.93)	0.20
MCHC (g/L)	339.00 (8.00)	329.50 (18.50)	0.15
RDW SD (fL)	41.70 (2.75)	42.75 (4.80)	0.34
RDW CV (%)	12.90 (0.80)	12.50 (1.22)	0.87
N (×10^9^/L)	3.91 (0.95)	3.72 (4.12)	0.75
L (×10^9^/L)	1.79 (0.55)	2.09 (1.32)	0.52
N/L	2.24 (0.84)	1.73 (3.44)	0.42
MON# (×10^9^/L)	0.52 (0.29)	0.44 (0.21)	0.87
EOS# (×10^9^/L)	0.10 (0.20)	0.24 (0.27)	0.08
Thrombocytocrit (%)	0.20 (0.04)	0.25 (0.07)	<0.01
PDW (fL)	11.70 (1.45)	11.60 (2.45)	0.58
MPV (fL)	10.35 (1.10)	10.20 (1.23)	0.47
PT (S)	11.35 (0.67)	11.25 (0.78)	0.69
APTT (S)	30.55 (4.85)	30.40 (4.17)	0.69
FIB (g/L)	2.70 (0.80)	3.56 (0.97)	0.04
Urea (mmol/L)	5.20 (1.05)	4.15 (2.25)	0.15
Cr (μmol/L)	85.00 (23.50)	65.00 (40.25)	0.57
UA (μmol/L)	442.00 (60.25)	302.00 (204.50)	0.34
K^+^ (mmol/L)	3.61 (0.44)	3.55 (0.29)	0.94
Na^+^ (mmol/L)	136.65 (1.88)	139.60 (3.05)	0.04
Cl^-^ (mmol/L)	100.45 (2.28)	100.95 (3.67)	0.52
GLU (mmol/L)	5.08 (1.17)	5.15 (0.64)	0.52
AST (U/L)	18.85 (21.18)	17.50 (22.92)	0.63
ALT (U/L)	16.50 (10.85)	20.80 (9.95)	0.42
TB (μmol/L)	13.50 (8.50)	7.95 (3.93)	0.06
DB (μmol/L)	5.25 (2.48)	3.20 (0.82)	0.03
Hemobilirubin (μmol/L)	8.45 (6.18)	4.75 (3.10)	0.07
TP (g/L)	69.00 (7.00)	72.25 (3.97)	0.26
Albumin (g/L)	42.20 (5.80)	43.55 (3.55)	0.87
Globulin (g/L)	25.75 (5.37)	29.00 (2.63)	0.13

Note: median (IQR), IQR, interquartile range. MPs: microplastics, WBC: white blood cell, RBC: red blood cell, HGB: hemoglobin, PLT: platelet, MCV: mean corpuscular volume, MCHC: mean corpuscular hemoglobin concentration, RDW: red blood cell volume distribution width, N: neutrophil, L: lymphocyte, MON#: monocyte, EOS#: eosinophil, PDW: platelet distribution width, MPV: mean platelet volume, PT: prothrombin time, APTT: activated partial thromboplastin time, FIB: fibrinogen, Cr: Creatinine, UA: blood uric acid, GLU: glucose, AST: aspartate aminotransferase, ALT: alanine aminotransferase, TB: total bilirubin, DB: direct bilirubin, TP: total protein. ^a^ MPs with a matching degree ≥ 0.80 were included. ^b^ The *p*-value was estimated using the Mann−Whitney U test.

## Data Availability

The data are not publicly available due to privacy or ethical considerations.

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
