# Peer review of "Microplastics in the Lung Tissues Associated with Blood Test Index"

_toxics, 2023, doi:10.3390/toxics11090759_

Round 1

Reviewer 1 Report

Review of: Microplastics in the lung tissues associated with blood test index

Toxics journal

Reference format in the main text of the manuscript can be consistently presented.

L43: Brahney et al., 2021 does not talk about traffic pollution. The citations have to be revised clearly in order to present them correctly and accordingly.

L51 – 53: please cite 10.1016/j.scitotenv.2022.159164

This is not the first study that has investigated the microplastics in the lungs from smokers and non-smokers. Authors have to check for recent articles in ASC ES & T journal. So that, the justification of the work can be improved at the end of introduction.

Section 2.1.

L116 – 117: What led the authors to believe that the proteins remained following nitric acid digestion?

L133 – 134: please cite: doi.org/10.1016/j.teac.2023.e00203

L148 – 149: recovery tests in replicates?

Blood index tests have very little predictive value, and it is impossible to link these results to the presence of microplastics. Even without it, the study's findings wouldn't have changed significantly, thus it could be left out.

The presentation of the results has to be improved. For example, the number of microplastics has to be expressed in terms of microplastics/g of lung tissue.

How were samples adjusted using blanks?

Fig. 1 present the full form for all the abbreviations used in the legend.

Fig. 2: what do red solid and blue dotted line indicate.

L221 – 225: it should be mentioned in the materials and methods, also show the questionnaire and

results in the supplementary material

L250 – 251: Please see above the comments.

Author Response

Reviewer 1:

Comment 1 (C1): L43: Brahney et al., 2021 does not talk about traffic pollution. The citations have to be revised clearly in order to present them correctly and accordingly.

Response 1 (R1): We have checked the citations and deleted it according to the reviewer’s suggestion.

C2: L51 – 53: please cite 10.1016/j.scitotenv.2022.159164.

R2: We thank for the reviewer’s suggestion. We have cited the related article in the revised manuscript (Page 2, line 51-53).

C3: This is not the first study that has investigated the MPs in the lungs from smokers and non-smokers. Authors have to check for recent articles in ASC ES & T journal. So that, the justification of the work can be improved at the end of introduction.

R3: We did not mention that this was the first study to investigate the MPs in the lungs from smokers and non-smokers. Actually, we excluded smokers and quantified MPs in lung tissues among never smokers. And to our knowledge, this is the first study to objectively and quantitively assess the associations between the MPs in the lung tissues and blood test index on human. We have revised the relative descriptions in the revised manuscript (Page 9, line 257-258)

C4: L116 – 117: What led the authors to believe that the proteins remained following nitric acid digestion?

R4: We used nitric acid to digest lung tissue and protein. Claessens et al., (Claessens et al., Mar Pollut Bull, 2013, 70:227–33) indicated that the digestion efficiency of the tissue can be achieved 95% by using nitric acid, but not 100%. To reduce the possibility of false positives, we excluded polyamide from our study because LDIR cannot distinguish polyamide and protein apart due to their similarity of spectrum.

C5: L133 – 134: please cite: doi.org/10.1016/j.teac.2023.e00203.

R5: We thank for the reviewer’s suggestion. We have cited the related article in the revised manuscript (Page 3, line 136-137).

C6: L148 – 149: recovery tests in replicates?

R6: We used three standard MPs samples including polystyrene (PS), polypropylene (PP) and polyethylene (PE) to calculate the recovery rate, which has been described in our previous study (Qiu et al., Environ Sci Technol, 2023, 57: 2435-2444). All experiments were repeated independently three times. The average recoveries of PS, PP, and PE were 92%, 89%, and 78% respectively, indicating the feasibility of LDIR. We have updated the related description in the revised manuscript (Page 4, line 152-153).

C7: The presentation of the results has to be improved. For example, the number of MPs has to be expressed in terms of MPs/g of lung tissue.

R7: The concentrations of MPs were calculated by dividing the quantity by the weight of the relevant lung tissue or control sample, and represented as particles/g. We have also added the unit of MPs in table 1 and table 2.

C8: How were samples adjusted using blanks?

R8: As described in the manuscript, we performed three blank control samples to cover all potential contamination sources, which were obtained by performing the same steps using 5.0 g ethanol instead of the human lung. The approach for the control group was substantially the same as for the lung tissue group, except that we utilized ethanol instead of human lungs. The opening time of glass bottles containing ethanol was the same as that of tissue samples. The operating rooms were previously cleaned, and conditions re-mained similar in each operating room during all sampling periods.

The blank controls were used to determine the presence of MPs in lung tissue by comparing the statistical differences between MPs in lung tissues and those in the blank controls. Which was widely used in human stool, placenta, lung tissue and sputum (Yan et al., Environ Sci Technol, 2022, 56: 414-421; Schwabl et al., Ann Intern Med, 2019, 171: 453-457; Braun et al., Pharmaceutics, 2021, 13: 921; Jenner et al., Sci Total Environ, 2022, 831: 154907; Huang et al., Environ Sci Technol, 2022, 56: 2476-2486).

C9: Fig. 1 present the full form for all the abbreviations used in the legend.

R9: Considering that the full names of various MPs are too long, we have given the full names of MPs in figure 1’s legend.

C10: Fig. 2: what do red solid and blue dotted line indicate.

R10: Red solid and blue dotted line respectively indicate the laser infrared spectrum of the sample and standard MPs. We have updated the related description of the figure 2 in the revised manuscript.

C11: L221 – 225: it should be mentioned in the materials and methods, also show the questionnaire and results in the supplementary material.

R11: We thank for the suggestion from the reviewer. We have updated the related descriptions in the revised manuscript (Page 2, line 74-76 and supplementary materials- table S2.).

C12: L250 – 251: Blood index tests have very little predictive value, and it is impossible to link these results to the presence of MPs. Even without it, the study's findings wouldn't have changed significantly, thus it could be left out.

R12: Blood test index is a frequent and essential indicator of human health. The relatively small sample size of this study limits the assessment of the dose-response relationship between MP exposure and blood test index levels, but the findings still provide some clues and data on the association between early health damage and exposure to MPs in the respiratory tract, which is not currently being investigated. We therefore believe that this finding has some scientific significance in revealing the toxic effects of human exposure to respirable MPs.

Reviewer 2 Report

See attached file.

Author Response

Reviewer 2:

The manuscript presented by Shuguang Wang et al. reports spectroscopic/imaging (LDIR) and SEM investigations of microplastics (MPs) found in human lung tissue. Samples were collected from 12 nonsmoking patients considering various factors e.g. sex, age, working indoors, alcohol consumption, traffic pollution etc. Authors found positive correlation between the level of MPs in lung tissues and platelets (PLT), thrombocytocrit, fibrinogen (FIB) and negative with direct bilirubin (DB).

In general, due to the increasing amount of microplastic particles/fibers in the environment (air, water) the topic of the manuscript is very important and interesting. In my opinion results are interesting and to best of my knowledge original and new.

However, some improvement of the manuscript should be carried out before publication in Toxics.

Please find enclosed questions, suggestions and comments:

Comment 1 (C1): Line 191 - 192 – what is the experimental evidence for weathering of the detected MPs?

Response 1 (R1): We appreciate the reviewer’s positive comments on our manuscript. The results of SEM showed that the surface of MPs were rough and irregular, and the LDIR results of MPs were different from those of standard MPs, which indicated that there might be weathering phenomena of MPs.

C2: In my opinion complementary method to those applied could be Raman microscopy with fast imaging. Using this method you can detect and identify microplastic particles/fibers. The spatial resolution depends on wavelength and NA of the optical objective and can be even below 1 μm. Modern systems allow to measure Raman maps even for macroscopic samples (in mm in size). What do you think about application of this technique in your study?

R2: Raman spectroscopy can indeed detect MPs with smaller particle sizes. It is a traditional optics method for MPs detection, which contains manual selection and is time-consuming. Compared with Raman spectroscopy, the LDIR is a high-throughput, automated technique for MPs detection, quantification, and characterization and has been widely used in a variety of biological samples including human placenta, meconium, semen, and sputum (Zhu et al., Sci Total Environ, 2022, 856:159060; Liu et al., Environ Sci Technol, 2023; Zhao et al., Sci Total Environ, 2023, 877:162713; Huang et al., Environ Sci Technol, 2022, 56:2476−86). It is less labor-intensive and time-consuming than Raman spectroscopy.

C3: Fig. 2 – please add a scale bar on the SEM images.

R3: We thank for the reviewer’s suggestion. We have updated the figure 2 in the revised manuscript.
